# Variation in a Darwin Wasp (Hymenoptera: Ichneumonidae) Community along an Elevation Gradient in a Tropical Biodiversity Hotspot: Implications for Ecology and Conservation

**DOI:** 10.3390/insects14110861

**Published:** 2023-11-07

**Authors:** Vivian Flinte, Diego G. Pádua, Emily M. Durand, Caitlin Hodgin, Gabriel Khattar, Luiz Felipe L. da Silveira, Daniell R. R. Fernandes, Ilari E. Sääksjärvi, Ricardo F. Monteiro, Margarete V. Macedo, Peter J. Mayhew

**Affiliations:** 1Departamento de Ecologia, Instituto de Biologia, Universidade Federal do Rio de Janeiro, C.P. 68020, Rio de Janeiro 21941-590, Brazil; vflinte@gmail.com (V.F.); gabriel.khattar@mail.concordia.ca (G.K.); silveira.lfl@gmail.com (L.F.L.d.S.); ricardomonteiroufrj@gmail.com (R.F.M.); margaretevmacedo@gmail.com (M.V.M.); 2Programa de Pós-Graduação em Entomologia, Instituto Nacional de Pesquisas da Amazônia, Manaus 69067-375, Brazil; paduadg@gmail.com (D.G.P.); daniellrrfernandes@gmail.com (D.R.R.F.); 3Centro de Investigación de Estudios Avanzados del Maule, Vicerrectoría de Investigación y Postgrado, Universidad Católica del Maule, Avenida San Miguel, Talca 3605, Chile; 4Department of Biology, University of York, Heslington, York YO10 5DD, UK; emilymaevad@gmail.com (E.M.D.); caitlinhodgin@hotmail.co.uk (C.H.); 5Laboratory of Community and Quantitative Ecology, Department of Biology, Concordia University, Montreal, QC H4B 1R6, Canada; 6Biology Department, Western Carolina University, 1 University Drive, Cullowhee, NC 28723, USA; 7Biodiversity Unit, University of Turku, 20014 Turku, Finland; ileesa@utu.fi

**Keywords:** altitudinal richness gradient, Brazilian Atlantic Rainforest, conservation strategy, latitudinal richness gradient, parasitoid wasp community, tropical insect diversity, biodiversity loss

## Abstract

**Simple Summary:**

How the diversity of species changes from place to place is well known for some groups, such as vertebrates, considerably aiding conservation planning. However, it is often poorly known for the very diverse groups that make up most species on Earth, such as many invertebrates. This may hinder their effective conservation. We surveyed a group of “Darwin wasps”, a very diverse and important component of invertebrate fauna, up a mountain in the Brazilian Atlantic Rainforest, a global hotspot for biodiversity. The wasps reproduce by parasitizing other insects and spiders. We found a large number of species of these wasps, suggesting that biodiversity may be high for this group in tropical regions, despite some previous studies suggesting the opposite. We found that low- and mid-altitude locations were especially diverse for this group, but not high-altitude locations. Furthermore, different species were found at different altitudes up the mountain. These findings suggest that tropical forests may harbor high concentrations of Darwin wasps; consequently, deforestation risks losing much of that biodiversity. Conserving low- to mid-altitude forests may be the most effective way to conserve the diversity of these wasps, although protecting a wide range of altitudes is necessary to conserve all species.

**Abstract:**

Understanding how biodiversity varies from place to place is a fundamental goal of ecology and an important tool for halting biodiversity loss. Parasitic wasps (Hymenoptera) are a diverse and functionally important animal group, but spatial variation in their diversity is poorly understood. We survey a community of parasitic wasps (Ichneumonidae: Pimplinae) using Malaise traps up a mountain in the Brazilian Atlantic Rainforest, and relate the catch to biotic and abiotic habitat characteristics. We find high species richness compared with previous similar studies, with abundance, richness, and diversity peaking at low to intermediate elevation. There is a marked change in community composition with elevation. Habitat factors strongly correlated with elevation also strongly predict changes in the pimpline community, including temperature as well as the density of bamboo, lianas, epiphytes, small trees, and herbs. These results identify several possible surrogates of pimpline communities in tropical forests, which could be used as a tool in conservation. They also contribute to the growing evidence for a typical latitudinal gradient in ichneumonid species richness, and suggest that low to medium elevations in tropical regions will sometimes conserve the greatest number of species locally, but to conserve maximal biodiversity, a wider range of elevations should also be targeted.

## 1. Introduction

Loss of Earth’s biodiversity is one of the most important issues facing humanity [1,2,3,4]. According to recent estimates, up to a million species risk extinction within the next few decades [5]. Mitigating such losses relies on a firm understanding of the ecological factors that contribute to variation in biodiversity [6]. Existing spatial variation in biological diversity is significant [7,8], and this variation can be a valuable source of information on the underlying factors that are likely to control it [9]. However, scientific effort related to the conservation of species is extremely biased, both spatially and taxonomically [10,11,12], and our understanding of these underlying factors is, unfortunately, best for relatively species-poor groups in relatively species-poor parts of the world [13,14]. In this paper, we document spatial community variation in a group of Ichneumonidae (Hymenoptera: Ichneumonoidea), also known as Darwin wasps [15], a highly diverse but understudied guild, in one of the most important global hotspots for biodiversity. We report how they are associated with abiotic and biotic habitat characteristics, with a view to enhancing our understanding of the factors affecting their diversity.

Anthropogenic activities are resulting in the serious loss of populations and species from areas of the planet that previously supported them [3,16]. The overall reasons for this decline are broadly known [5], including land use change, leading to destruction, degradation, and fragmentation of natural habitats [17]; pollution, including fossil-fuel-induced climate change [18]; the introduction of invasive species to new localities by humans [19]; and overexploitation of commercially valuable species [20]. A major tool to help mitigate biodiversity loss is an understanding of the biotic and abiotic conditions, and corresponding places that favor high, or conversely encourage low, diversity [21,22,23]. For example, such knowledge could enable policy makers to create and manage protected areas to maximize the number of species conserved [24].

However, a potentially serious impediment to effective conservation is knowledge bias [10,11,12]. Data on species distributions are most abundant in certain temperate localities (particularly Europe and North America), whilst biological diversity is usually maximal in the tropics [10]. Similarly, the taxa that are most studied by conservation biologists are vertebrates; yet, biological diversity is highest in other taxa, such as invertebrates [11]. Consequently, conservation strategies based on an understanding of well-known taxa may be sub-optimal for the vast majority of Earth’s species [12]. One way to solve this predicament is to improve our knowledge of the distribution, ecology, and conservation of highly diverse but less-studied taxa.

Parasitoid wasps (Hymenoptera) are a group of insects with about 80,000 described species [25], thus representing less than 5% of the total species described to date [26]. However, for a variety of reasons (e.g., dearth of taxonomic specialists, cryptic species, lack of long-term field research in the tropics), this total is likely to be largely underestimated, with one study suggesting true species richness to be ten times the described species richness, making these wasps a very significant component of macroscopic biodiversity [27]. The larvae of parasitoid wasps feed on the still-living bodies of other arthropods, normally other insects, eventually killing them. However, the adults are free-living, with the females searching for new hosts upon which, or in which, to lay their eggs [28,29]. Because their life cycle results in the death of their hosts, they play an important ecological function by regulating the populations of other arthropods, including pests, and are widely used as biocontrol agents [30,31]. In common with many such diverse groups, parasitoid wasps are rarely explicitly incorporated into conservation strategies [32,33]. Intuitively, their specialized biology and high trophic level, together with the sparse existing data [33,34,35], suggests that they may be particularly susceptible to anthropogenic threats.

One way in which parasitoids might reasonably be incorporated into conservation strategies is if other features of the environment that are better known or more easily measured (these could be abiotic or biotic) could serve as surrogates [36,37]. One basic and simple potential surrogate over large spatial scales is the latitude of a site [21], as in most taxa there is a strong latitudinal species richness gradient, with richness being higher at the equator than at the poles [38,39]. Interestingly though, even the basic latitudinal richness pattern is not established with confidence in parasitoid wasps [40]. It has, for example, been claimed that for one diverse family, the Darwin wasps, the latitudinal gradient may even be “inverse” (actually modal) [41,42], with more species in temperate regions than tropical regions [40]. However, recent work suggests that this supposed pattern might be partly the result of incomplete sampling [43], as more exhaustive tropical sampling [44,45,46], and sampling that identifies cryptic species [47,48], has produced higher estimates for tropical samples.

Alongside latitude, elevation (altitude) is another environmental factor that can strongly affect species richness [49,50]. Across taxa, elevational species richness gradients typically show low richness at the very highest elevations [51]. Many studies show that richness increases as elevation reduces, but other studies show modal patterns where richness is highest at intermediate elevations [52]. In parasitoids, some studies show such modal patterns [53,54,55], others show no elevational trend [56], and yet others show highest richness or diversity at high elevations [57,58,59]. Within studies, different taxonomic groups can show different elevational patterns [55,60]. As well as understanding standing diversity, the turnover of diversity with elevation is also important for conservation planning, because protected areas are likely to extend over a range of elevations. Again, existing data suggest that parasitoid wasp species [61,62], or higher taxa [55,63] may occupy a limited range of elevations. This means that protecting a range of elevations can be beneficial through the incorporation of more local suites of species.

Other biotic factors may predict parasitoid richness at a more local scale. Within regions, parasitoid communities often vary according to local vegetation type [45,64,65,66,67]. However, vegetation type may not always affect the diversity of all parasitoid groups [56,65]. Several studies suggest that habitat structural complexity can predict parasitoid wasp diversity in both natural and agroecosystems [57,68,69,70,71]. Plant diversity can also predict wasp diversity across sites [64,65,66,72,73,74,75,76]. For example, Fraser et al. [65] showed that tree species’ richness was a good predictor of species richness in pimpline ichneumonids living in temperate woodlands. The authors also showed that applying a nature reserve selection algorithm, this surrogate (tree richness) could provide almost as good protection for pimplines as knowledge of the pimpline distribution itself [77]. However, it is still unclear to what extent vegetation characteristics such as these may be effective in prioritizing parasitoid conservation, especially in diverse tropical ecosystems.

In this study, we survey variation in a community of the Darwin wasp subfamily, Pimplinae, along an elevational gradient up a mountain in the Brazilian Atlantic Rainforest, a tropical biodiversity hotspot. We compare the observed richness of our collections with that of other similar collections of the same group around the world, controlling for elevation and sampling effort. We find that richness is high in our collections, consistent with the existence of a typical latitudinal richness gradient for the group. We then observe how diversity (alpha and beta) varies with elevation. We find that low to mid elevations contain the highest diversity locally, but turnover of species up the mountain is strong, which might impact the best conservation strategies. Finally, we identify the biotic and abiotic variables that best predict abundance, richness, and diversity, which could represent potential conservation surrogates for this group.

## 2. Materials and Methods

### 2.1. Study Site

We collected pimpline wasps at Serra dos Órgãos National Park in Rio de Janeiro State, Brazil (22°32′ S and 43°07′ W) (Figure 1), which is part of the Atlantic Rainforest. The Atlantic Rainforest biome has been designated as one of the most important global biodiversity hotspots [78,79]. It is distributed across a heterogeneous landscape that includes ombrophilous forests, semi-deciduous forests, mountain cloud forests, campos de altitude (high-altitude grasslands), inselbergs, restingas (dune systems), and mangroves [80,81]. Currently, estimates of native Atlantic Forest cover range from ~11% [80] to ~28% [82]. However, 80% of the remnant forested area comprises patches of less than 50 ha [80]. Due to the significant changes in the biome over past decades, hundreds of animal and plant species are now at risk of extinction [83]. Some 40% of its vascular plants and 60% of vertebrates are endemic species [78]. The insect fauna of the biome is relatively poorly known, and it is still easy to find many new species (e.g., [84]).

The Serra dos Órgãos National Park contains the highest peaks of the Serra do Mar, just inland from the Atlantic coast, and the land ranges in elevation from 80 m to 2263 m. Founded in 1939, it is the third oldest National Park in Brazil. The region is among the best preserved in the Atlantic Forest biome [85], although much of the forest is secondary growth. The Park encloses four different vegetation belts: lower montane forest (below ~800 m), montane forest (800–1500 m), upper montane forest (1500–2000 m), and high-elevation grasslands, campos de altitude (over ~2000 m), characterized by shrubs, herbs, and grasses [86,87], with the latter comprising a large fraction of the botanical endemism. The mountains are characterized by decreasing temperature with increasing elevation, about 0.5 °C per 100 m elevation [62,88], with mean monthly temperatures varying from about 22 °C at the base to 12 °C on the mountain peaks. There is a season of higher precipitation and temperatures (the rainy season), from October to March, and relatively a drier and cooler season (the dry season), from April to September [85].

Pimplinae wasps were collected along a transect from 110 m to 2169 m elevation (these being our lowest and highest elevation trap locations, see Appendix A) up a mountain in the National Park [88]. The transect first followed the route of Highway BR-116 from Guapimirim to Teresópolis, and thereafter, from the Park entrance in Teresópolis along the Park main road and the Pedro do Sino trail (Figure 1).

### 2.2. Study Species

As the family of Darwin wasps is extremely diverse and encompasses a large variety of different parasitoid insect life history strategies [89], it is most convenient to study the altitudinal and latitudinal species richness gradient with smaller clades within the family. Our study focused on parasitoid wasps of the family Ichneumonidae, subfamily Pimplinae. The Ichneumonidae, or Darwin wasps, is one of the most species-rich families of Hymenoptera, with more than 25,000 described species in over 1600 genera divided into 41 subfamilies [90,91]. The Pimplinae is one of the richest of these subfamilies, with about 1700 described species in about 79 genera [92]. It is biologically diverse and associated with a wide range of hosts [89]. Species are mainly idiobiont ectoparasitoids of the immature stages of holometabolan insects or idiobiont endoparasitoids of lepidopteran or hymenopteran pupae. However, species of the *Polysphincta* genus group are koinobiont ectoparasitoids of spiders [93]. This biological and taxonomic diversity, together with relatively good taxonomic knowledge, has made them a frequent target for ecological studies of parasitoid communities.

### 2.3. Pimplinae Collections

We collected pimplines using 30 Malaise traps (a form a flight interception trap) distributed at 15 broad elevation sites throughout the transect, with two replicate traps at each site. Malaise traps are one of the most efficient methods for sampling Ichneumonidae and are widely used in ecological studies of insects [94]. Traps were set at ground level, with the collecting head 1.5 m above the ground. Traps were placed at least 50 m from the road and the 15 sites were roughly spaced at 100 m to 200 m elevation intervals. At each elevational site, the two traps were placed at least 50 m apart from each other to ensure that neither trap affected the catch of the other and to sample different environmental space [65]. Trap collecting bottles (1 L capacity) contained 98% ethanol for preservation of the sampled material and were replaced monthly (after 30 days of collecting). The samples were collected during both the rainy hot season, from December 2014 to February 2015, and the dry cooler season from June to August 2015, totaling 180 Malaise trap months. These months were chosen to represent the opposite environmental extremes throughout the year to capture seasonal variations in species composition, but also to encompass the warmest, wettest months when insect activity is expected to be highest (December–February).

Insects were preserved in 98% ethanol and stored in plastic bottles. Sample sorting was performed in the laboratory using a stereoscopic microscope. Identification was initially carried out according to subfamily by D.R.R.F. and D.G.P. following [92]. Then, the Pimplinae were identified according to genus following [89]. Thereafter, morphospecies were identified by D.G.P. in conjunction with I.E.S., and where possible, species (using specific bibliography and large reference collections of neotropical Darwin wasps in the Biodiversity Unit, University of Turku, Finland). All the researchers involved in identification were experienced in neotropical ichneumonid taxonomy. The use of morphospecies (i.e., individuals sorted based on phenotypic characteristics) as surrogates for species is widely used in the estimation of species richness for comparisons over time and space [95,96]. Although the designation of morphospecies can lead to the split of a single species into many different morphospecies (“splitting”) or aggregation of different species into a single species (“lumping”), it is often the only way to assess species diversity in groups that have not been fully described [95,96].

Collections were performed under license number 21409-10 (Ministério do Meio Ambiente—MMA; Instituto Chico Mendes de Conservação da Biodiversidade—ICMBio; Sistema de Autorização e Informação em Biodiversidade—SISBIO) to Ricardo Ferreira Monteiro.

The sampled material is deposited at the following Brazilian entomological collections: Invertebrate Collection of Instituto Nacional de Pesquisas da Amazônia, Manaus, Brazil (INPA), (curator: Marcio L. Oliveira); Museu de Zoologia da Universidade de São Paulo, São Paulo, Brazil (MZUSP), (curator: Gabriela P. Camacho); and Taxonomic Collection of the Departamento de Ecologia e Biologia Evolutiva from Universidade Federal de São Carlos, São Carlos, Brazil (DCBU), (curator: Angelica M. Penteado-Dias).

### 2.4. Environmental Variables

Hand-held GPS units were used to find the latitude and longitude of each trap on the ground, from which elevation was estimated by inputting those locations into the digital elevation model in Google Earth.

Temperature (°C) and relative humidity data were collected every hour for all 15 sampling sites using automatic Data Loggers (MicroLite II USB Temperature Data Logger, Fourtec—Fourier Technologies Ltd., Rosh Haayin, Israel) for a full year from December 2014 to November 2015, thus representing a complete annual record of each site during the sampling year. From these data, we obtained the mean, maximum, and minimum daily temperature and humidity. From this, we calculated the monthly means for each elevational site. Temperature amplitude (the difference between the monthly minima and maxima for each site) was also calculated. The full annual record was used since species composition is likely affected by environmental conditions over the full range of the seasons.

Other habitat variables were collected at all 30 traps separately on 1 and 2 September 2015. Three replicate samples of each variable were taken next to each trap and the mean of the three samples taken. All leaf litter within a 20 cm diameter circle was collected and weighed on a microbalance both fresh and after drying. Litter was dried in an oven at 60 °C for at least 48 h or when the weight stopped declining. The difference between fresh and dry mass was calculated to determine the mass of litter moisture. The percentage of litter moisture was calculated as (mass litter moisture/litter dry mass) × 100. The density of larger trees, with DBH ≥ 10 cm was counted in 10 m × 2 m areas next to each trap. The number of these trees supporting lianas and epiphytes was also counted. The density of smaller trees (DBH < 10 cm) was counted in 5 m × 2 m areas next to each trap, and again the number of these trees supporting lianas and epiphytes was also counted. The total density of plants with lianas and epiphytes was then calculated by adding the counts for larger trees to 2× the count for smaller trees. Bamboo and fern ground cover were each assessed in 5 m × 2 m areas around each trap on an ordinal scale of 0 to 3, with 0 being absent, 1 being up to 25%, 2 between 25% and 75%, and 3 being over 75%. Herb ground cover was assessed using 0.5 m × 0.5 m quadrats, and scored based on photos taken on site on an ordinal scale of 0 to 4, with 0 being absent, 1 being up to 25%, 2 being between 25% and 50%, 3 being between 50% and 75%, and 4 being over 75%. These different assessment procedures were conducted to facilitate more accurate and rapid assessment of each type of vegetation following initial testing using different recorders. The density of palms and tree ferns was counted in 5 m × 2 m areas next to each trap.

### 2.5. Analysis

Analysis was carried out at both site level (*n* = 15) and trap level (*n* = 30) where data allowed, although unsurprisingly, findings were generally highly congruent for these different analyses. Because the temperature and humidity data were not recorded at the scale of individual traps (only one data logger per site), those variables were only included in site level analyses.

Two diversity indices were calculated for the pimpline community using the function diversity in the vegan package [97] in R version 4.2.3 [98] to test the sensitivity of the conclusions related to choice of index: the inverse (1/*D*) Simpson’s and Shannon. Simpson’s is recommended when communities are incompletely sampled and gives more weight to common species and evenness, whilst Shannon is more heavily influenced by rare species and species richness [99]. In addition to these diversity indices, we summed the total number of individuals and number of morphospecies (species richness) to use as response variables.

To assess how well we had sampled the pimpline community, we constructed species accumulation and rarefaction curves using the function specaccum in the vegan package in R. Estimates of the total number of species in the community were made using both incidence (Chao, first-order jackknife, second-order jackknife, bootstrap [100]) and abundance (bias-corrected Chao and ACE, [101,102]) based estimators (±SE), using the function poolaccum in vegan. The bias-corrected Chao and ACE estimators were based on abundance across all samples combined.

To compare our observed species richness with other comparable samples from around the world, we used the large sample of other pimpline Malaise collections collated from the published literature in [45]. Some of these samples also included Rhyssinae (once treated as a tribe of Pimplinae but now a separate subfamily not included in our study), but these only ever contained a very small proportion of individuals and species sampled. All of these other samples were taken from a much smaller range of elevations (<200 m) than our study; so, to make our data more comparable, we split them into seven zones by elevation (each zone comprising two of the collection sites except the highest zone, which comprised three) and analyzed them separately, plotting rarefaction curves for each to compare with the literature data.

To test whether and how pimpline abundance, richness, and diversity varied with elevation, we constructed linear models of each variable against elevation, and then compared this against models including elevation^2^ (a quadratic model giving a simple curve), elevation^3^ (a cubic model allowing sigmoidal curves), and elevation^4^ (a quartic model allowing more complex curvature). We compared the models using AICc, with the preferred models having lower AICc scores. The appropriateness of the models (distribution of residuals, influence of outliers) was assessed using the plot function. In this case, residual plots showed that species richness and inverse Simpson’s diversity produced poor model fits. This was solved by transforming them before analysis using log_10_. These analyses were also repeated at the site level for mean monthly temperature instead of elevation, since elevation and temperature were highly correlated, and since temperature might be an interesting practical conservation surrogate.

To investigate beta diversity (turnover) with elevation, we calculated pairwise Bray–Curtis dissimilarity indices of sites and traps using the vegdist function in vegan. To assess whether community dissimilarity was correlated with elevation distance, we first constructed a matrix of pairwise Euclidean distances between traps and sites for elevation (using the dist function), and then used a Mantel test to correlate it against Bray–Curtis (using the mantel function) indices using 9999 permutations and the Spearman method.

To explore likely predictors of the various community metrics and to assess redundant or co-linear habitat variables, a matrix of Spearman’s correlation coefficients was examined between all variables, with significance adjusted for multiple comparison using the false discovery rate correction [103]. Following this, Principal Component Analysis (PCA) was used to reduce the habitat variables to a smaller number of orthogonal (independent) variables using the prcomp function in R. The data were standardized before analysis using the scale = TRUE option. We identified significant components as those with initial eigenvalues > 1. These principal components (PCs) were then used as predictor variables of pimpline abundance, richness, and diversity in linear models, with alternative models assessed through multi-model selection via AICc [104] using the function dredge in the R package MuMin [105].

To explore how community composition varied across traps and sites, community ordination was carried out using Non-metric Multidimensional Scaling (NMDS) on Bray–Curtis dissimilarity, using the metaMDS function in the R package vegan. To test whether habitat PCs could predict pimpline community composition, site, or trap scores from the NMDS axes were used as response variables in general linear models using the habitat principal components as predictor variables, again using multi-model inference to assess models via AICc.

To assess which of the vegetation-based habitat variables best predicted pimpline abundance, richness, diversity, and species composition (NMDS site or trap scores), we conducted two types of linear model analysis. In the first, all models were constrained first to contain elevation, or where appropriate, quadratic or cubic elevation terms as well. These models tested whether any vegetation features could predict the pimpline community above and beyond the elevation, assuming that the effect of elevation was a “nuisance” factor to be removed and did not exert its effects primarily via vegetation. A second set of models only included vegetation variables, assuming these to be the primary means through which elevation effects are exerted, and testing whether these could be used as direct surrogates of the pimpline community across a wide range of elevations. Again, models were assessed via multi-model selection using AICc.

## 3. Results

### 3.1. Species Richness

In the six months of sorted pimpline collections, we found 1560 pimpline individuals in 19 genera and 98 morphospecies (Figure 2), of which 24 species were ascribed to described species (could be named by us). The most common species was *Pimpla caerulea* Brullé with 447 individuals (Figure 2), while the second most abundant was *Pimpla croceiventris* (Cresson) with 76 individuals. In contrast, 25 species were represented in the sample by just a single individual (singletons), and 11 species by just two individuals (doubletons).

Species accumulation and rarefaction curves suggested that the community was incompletely sampled (Figure 3) with respect to our sampling method, dates, and locations; however, the curves, whether scoring samples by traps (Figure 3a,c), sites (Figure 3b), or individuals (Figure 3d), appeared to be gradually asymptoting. Estimates of the total species pool suggested that 69–89% of the species pool were actually in our samples (Table 1). The estimated total species pool rose with sampling intensity (Figure 3a), and the estimated number of unsampled species remained relatively consistent (Figure 3a), suggesting that the true species pool could be larger than the estimators suggest.

The rarefaction curve for the whole dataset (trap level) suggests that species richness is high compared with other pimpline collections using Malaise traps published in the literature (Figure 3d). The only two collections with higher richness are from lowland Peruvian Amazonia or the Peruvian Andean–Amazonian interface [45] and slightly exceed those seen here for a given number of individuals. In addition, another recently published collection from SE Brazil at three mid-elevation altitudes reports more species (91 in total) than expected from our samples, given the number of individuals sampled (745) [63].

The results for the different altitudinal zones considered separately are slightly more mixed. The curves for the two highest-altitude zones fall within the normal range of species expected in the literature for a given sample size (Figure 3d). However, the zones from the lower and mid elevations show a high richness for a given number of individuals compared with previous studies. Previous collections that have reported equivalent richness for a given number of individuals are all tropical, and include Boqueirão, Brazil [106], several sites in Costa Rica [107], and Langat Basin, Malaysia [108].

### 3.2. Community Variation with Elevation and Temperature

Linear models suggest that the high-elevation traps produced the lowest pimpline abundance, richness, and diversity, though the precise relationship varied across the different community metrics. For abundance, the best elevation model was cubic (Table 2), with a peak at about 1500 m (Figure 4a). For (log of) species richness, the best model was cubic (Table 2), with high richness below 1400 m (Figure 4b). For the (log of) inverse Simpson’s Index, the model with the lowest AICc was quadratic, but a linear model was within two AICc units of this (Table 2). Thus, Simpson’s diversity was highest below 700 m (Figure 4c). For Shannon diversity, a quadratic model had the lowest AICc, but a cubic model was almost as good (Table 2). Shannon diversity was highest at 600 m, only dropping at high elevation (Figure 4d). Unsurprisingly, the results at site level were very similar (Appendix A).

At the site level, a cubic model of mean monthly temperatures gave a peak abundance at about 16 °C, log richness (in a quadratic model) peaked at about 19 °C, the highest (log_10_) inverse Simpson’s diversity was at the highest temperatures (22.4 °C), whilst Shannon diversity peaked at about 20 °C (Figure 5 and Table 3).

Community dissimilarity (as measured via the Bray–Curtis index) was strongly correlated with elevation differences across sites (Mantel’s *r* = 0.842, *p* = 0.0001). The lowest dissimilarity was between sites 11 and 12, high to medium elevation, (BCI = 0.273) and adjacent sites tended to have low dissimilarity, whereas several pairs of sites had maximal dissimilarity (BCI = 1), indicating no overlap in species (Appendix A). These pairs included sites 1 and 14; 1 and 15; 2 and 15 (i.e., pairs containing one high elevation and one low elevation site) as well as 5 and 14 and 5 and 15. The latter result is explained through site 5 containing relatively few species for its elevation, which increases the chance of species not being shared with other sites.

The results were very similar at trap level, again with very strong correlation between community dissimilarity and elevation differences (Mantel’s *r* = 0.760, *p* = 0.0001). The lowest dissimilarity was between two traps at the same site at high altitude (traps 14A and 14B, BCI = 0.176), while dissimilarity was generally lowest between traps at the same site or traps at adjacent sites (Appendix A). Unsurprisingly, there were several pairs of traps with no overlap in species (BCI = 1). These tended to be pairs containing one high elevation and one low elevation trap; for example, trap 1A had no overlap of species with traps 13B, 14A, 14B, 15A, and 15B, but also had no overlap with trap 5A.

### 3.3. Associations between Pimpline Community and Other Habitat Properties

To explore likely predictors of the various community metrics, and to assess redundant or co-linear habitat variables, a matrix of Spearman’s correlation coefficients was examined between all variables (Appendix A).

At site level (*n* = 15), some of the strongest correlations were between the different pimpline community metrics (abundance, richness, and diversity), which were unsurprisingly often positively correlated with each other (Appendix A). Abundance was not strongly correlated with any other habitat variable (although as described above for elevation, it may show non-linear relationships with some variables). Richness was negatively correlated with herb ground cover. The inverse Simpson’s Index was positively correlated with mean, maximum, and minimum temperature, and negatively correlated with elevation, temperature amplitude, % litter moisture, and the cover of bamboo, ferns, herbs, and epiphyte density. Shannon diversity showed the same broad correlations.

For the abiotic variables, elevation was negatively correlated with all temperature measures except amplitude (which was positively correlated). It was positively correlated with % of litter moisture, bamboo, and fern ground cover as well as epiphyte density (Appendix A). Mean, maximum, and minimum temperature showed largely the same set of associations as elevation, only reversed, and were very strongly negatively associated with elevation. Temperature amplitude showed relationships with other variables, consistent with a positive association with elevation. Humidity was almost always nearly 100% and showed very little variation or association with any other variables. Dry litter mass was negatively correlated with herb ground cover. The percentage of litter moisture was positively correlated with elevation and temperature amplitude, bamboo and fern ground cover, and epiphyte density. It was negatively correlated with mean, maximum, and minimum temperatures.

For the vegetation variables, the density of large trees was not significantly correlated with any other habitat variable, and neither was the density of small trees or tree ferns. The density of palms was positively correlated with liana density and negatively correlated with epiphyte density (Appendix A). Bamboo cover was positively correlated with elevation, temperature amplitude, litter moisture, fern cover, and epiphyte density, and negatively correlated with mean, maximum, and minimum temperature and liana density. Fern ground cover showed similar associations. Herb ground cover was positively correlated with fern ground cover, and negatively with litter mass. Epiphyte density was positively correlated with elevation, temperature amplitude, % litter moisture, bamboo and fern ground cover, and negatively with mean, maximum, and minimum temperature as well as palm density and liana density. Liana density was positively correlated with palm density and negatively with bamboo cover and epiphyte density.

Overall, these associations are consistent with a picture in which several variables show strong associations with elevation, and therefore, each other. Most of the associations of habitat variables with the pimpline community follow from these elevational associations. Other variables show little association with elevation but are associated with some other habitat variables, for example, liana with palm density (negatively).

The correlations were very similar at trap level for those variables recorded at that level (Appendix A). For example, elevation was positively correlated with litter moisture, bamboo cover, fern cover, and epiphyte density, as before. However, richness was now positively correlated with palm density as well as herb cover. Inverse Simpson’s and Shannon diversity were also positively correlated with palm density. Small tree density was positively correlated with liana density.

A PCA on the site-level habitat variables produced four PCs with initial eigenvalues > 1 (Table 4). The first component explained 49.14% of the variance and was positively weighted by mean, maximum, and minimum temperatures, and negatively weighted by elevation, litter moisture, bamboo and fern ground cover, and epiphyte density. The second component (15.63% variance explained) was positively weighted by herb ground cover and negatively by large tree density and dry litter mass. The third component (10.61% variance explained) was negatively weighted by humidity, large and small tree and palm density. The fourth component (8.69% variance explained) was negatively weighted by tree fern density and humidity, and positively by small tree density and herb ground cover.

Multi-model inference using the first four PC axis site scores gave the following (Table 5): for abundance, the only model within two AICc units of the best (null) model was that including PC3, which was not significant. However, a quadratic model of PC1 was significant and had a lower AICc than either the linear or cubic model (Table 5). For (log_10_) richness, the best model included only PC1, which was positively associated with richness. The only other model within two AICc units of the best (null) model was that including both PC1 and PC3, but PC3 was not significant. A quadratic model of PC1 provided a better residual distribution and lower AICc score (Table 5). For (log_10_) inverse Simpson’s Index, the best model contained PC1, and the only other model within two AICc units of the best contained both PC1 and PC2, but PC2 was not significant. PC1 was positively associated with inverse Simpson’s Index. For the Shannon Index, again, the best model contained only PC1, and only one other model was within AICc units, containing both PC1 and PC2, but the latter was not significant. Again, PC1 was positively associated with the Shannon Index.

NMDS using the Bray–Curtis dissimilarity index strongly suggested that sites at higher elevation tended to have higher NMDS1 scores (Figure 6). Species with high scores on NMDS1 included *Calliephialtes* sp. 6 (1.79), *Calliephialtes* sp. 1 (1.59), *Eruga* sp. 2 (1.35), and *Clistopyga* sp. 3 (1.30). Species with low NMDS1 scores included *Zatypota* sp. 3 (−1.48), *Neotheronia lineata* (−1.44), *Neotheronia* sp. 14 (−1.41), and *Neotheronia* sp. 13 (−1.37).

In multi-model inference, the best model of NMDS1 contained both PC1 (Figure 7) and PC2, and was negatively associated with both (although PC2 was marginally non-significant), explaining 94.65% of the variance in NMDS1 (*b*(PC1) = 0.294, *t*_12_ = −14.40, *p* < 0.001; *b*(PC2) = 0.0783, *t*_12_ = −2.16, *p* = 0.051). NMDS2, however, was best explained by a null model, and none of the PC axes was significant.

At the trap level, a PCA on the habitat variables produced very similar results to the PCA at site level (Appendix A). Four components had initial eigenvalues > 1. A first PC explaining 34.44% of the variance was weighted positively with elevation, litter moisture, bamboo and fern ground cover, and epiphyte density. A second PC explained 16.88% of the variance and was weighted positively with herb ground cover, and negatively with litter mass and large and small tree density. A third component explained 13.97% of the variance and was negatively weighted with small tree, tree fern, and liana density. A fourth component explained 8.98% of the variance. It was positively weighted with palm density and herb ground cover, and negatively with tree fern density.

In multi-model selection, for abundance, the best model was null, and no PC axes were significant predictors, although a quadratic model containing PC1 was significant (Appendix A). For (log_10_) richness, the best model contained PC1 and was the only significant predictor, showing a negative association. A quadratic model of PC1 provided a better residual distribution and lower AICc score (Appendix A). For the (log_10_) inverse Simpson’s Index, the best model contained both PC1 and PC2; both were negatively associated, although PC2 was not significant. For the Shannon Index, the best model contained only PC1 (Appendix A).

An NMDS on the trap level community again produced a first axis that was clearly related to elevation (Appendix A). Species with high scores on this axis included *Pimpla caerulea* (1.38), *Calliephialtes* sp. 6 (1.26), *Calliephialtes* sp. 1 (1.12), and *Clistopyga* sp. 1 (1.10). Species with high negative scores included *Zatypota* sp. 3 (−1.46), *Neotheronia* sp. 14 (−1.33), *Neotheronia* sp. 13 (−1.27), and *Neotheronia* sp. 16 (−1.26).

The best model explaining NMDS1 included PC1, PC3, and PC4, all of which were significant (PC1: *t*_26_ = 12.79, *p* < 0.001; PC3: *t*_26_ = 2.665, *p* = 0.0131; PC4: *t*_26_ = −2.431, *p* = 0.0223, model *r*^2^ = 0.872). PC1 was positively associated with NMDS1 (*b* = 0.353, Appendix A), as was PC3 (*b* = 0.116), but PC4 was negatively associated with NMDS1 (*b* = −1.32). The best model of NMDS2 was null, and none of the PCs was significantly associated with it.

### 3.4. Vegetation Predictors of the Pimpline Community

Across 15 sites, when multi-model selection was used on abundance, constraining all models to contain cubic (plus squared plus linear) elevation terms, the best model contained just those terms and no vegetation variables, meaning that no vegetation variables explained abundance beyond elevation itself. For (log of) richness, models were constrained to contain quadratic and linear elevation terms. Again, the best model contained just these terms and two models within two AICc units additionally contained fern ground cover (*b* = −0.217, *t*_11_ = −1.83, *p* = 0.095) and herb ground cover (*b* = −0.164, *t*_11_ = −1.56, *p* = 0.147), both negatively but not significantly associated with log richness. For (log_10_) inverse Simpson’s Index, models were constrained to contain elevation. The best model, and only one within two AICc units, contained herb ground cover, which was negatively associated with diversity (*b* = −0.321, *t*_12_ = −3.61, *p* = 0.004). The Shannon diversity models were constrained to include elevation and elevation squared. The best model again contained herb ground cover (*b* = −0.645, *t*_11_ = −3.46, *p* = 0.005). For NMDS1, the best and only model contained elevation (*b* = 0.001, *t*_12_ = 10.66, *p* < 0.001) and epiphyte density (*b* = 0.0548, *t*_12_ = 3.004, *p* = 0.011), which were both positively and significantly associated with NMDS1. These results suggest that few variables predict the pimpline community in our study beyond those that might be associated with elevation.

In models without elevation and only vegetation variables at the site scale, for abundance, the suite of four best models (within two AICc units) contained liana and epiphyte density and herb ground cover, but none was significant (conditional model averages: lianas *b* = 14.62, *p* = 0.263; epiphytes *b* = −4.852, *p* = 0.263, herbs *b* = −30.04, *p* = 0.325). For log richness, the same three variables were in the best suite of six models, and two were conditionally significant: log richness was positively associated with liana density (conditional model average *b* = 0.123, *p* = 0.0194), negatively with epiphyte density (conditional model average *b* = −0.0568, *p* = 0.00693). Herb cover was not significant (conditional model average *b* = −0.275, *p* = 0.0503), and non-significant terms also included were large tree density (conditional model average *b* = 0.0916, *p* = 0.256) and small tree density (conditional model average *b* = 0.0381, *p* = 0.252). The best model contained epiphyte density alone. For (log_10_) inverse Simpson’s Index, the best and only model (*r*^2^ = 0.755) contained bamboo cover (*b* = −0.224, *t*_12_ = −3.959, *p* = 0.0019) and herb cover (*b* = −0.274, *t*_12_ = −2.583, *p* = 0.0240), both negatively associated with diversity. For Shannon diversity, the best suite of five models contained herb cover, small tree density, epiphyte density, and bamboo cover. Small tree density was positively associated with diversity (conditional model average *b* = 0.117, *p* = 0.0736) and epiphyte density (conditional model average *b* = −0.159, *p* = 0.00009), herb cover (conditional model average *b* = −0.530, *p* = 0.0351), and bamboo cover (conditional model average *b* = −0.450, *p* = 0.0004) negatively. For NMDS1, bamboo cover was the only variable in the best model and the only significant predictor (conditional model average *b* = 0.727, *p* = 0.0002), being positively associated with NMDS1.

At the trap scale, for models including elevation, abundance models were constrained to contain a cubic, quadratic, and linear elevation term. The best suite of (2) models (within two AICc units) also only contained small tree density, but this was not significant (conditional model average *b* = 1.90, *p* = 0.282). For log richness, with models constrained to contain quadratic and linear elevation terms, the only other significant variable in the best model suite was epiphyte density (conditional model average *b* = −0.0313, *p* = 0.0410). For the (log_10_) inverse Simpson’s Index, the best models constrained to contain quadratic and linear elevation terms also contained epiphyte density (conditional model average *b* = −0.029, *p* = 0.0375) and herb cover (conditional model average *b* = −0.195, *p* = 0.005), which were significantly negatively associated with diversity, and small tree density (conditional model average *b* = 0.0240, *p* = 0.217), which was positively but not significantly associated with diversity. Very similar results were found for Shannon diversity (conditional model averages: epiphytes *b* = −0.0692, *p* = 0.015; herbs *b* = −0.385, *p* = 0.014; small trees *b* = 0.0473, *p* = 0.275), although in addition, the best model suite also contained bamboo cover (conditional model average *b* = 0.200, *p* = 0.1720), which was positively but not significantly associated with diversity. For models of NMDS1 constrained to include elevation, the suite of best models contained large tree, epiphyte, and palm density as well as herb cover, of which only epiphyte density was significant (conditional model average *b* = 0.0285, *p* = 0.00225), being positively associated with NMDS1.

For models without elevation and with just vegetation variables, the best abundance model was null, and the only other variables in the best model suite (within two AICc units) were herb cover and epiphyte density, but these were not significant. For log richness, the best models contained herb cover as well as epiphyte, small tree, large tree, and liana density, with the former two positively and the others negatively associated with richness, but only epiphyte density was significant (conditional model average *b* = −0.0533, *p* = 0.0001) (Figure 8a). For the (log_10_) of the inverse Simpson’s Index, the single best model contained small tree density (*b* = 0.0405, *t*_26_ = 2.256, *p* = 0.0327), a positive association, herb cover (*b* = −0.189, *t*_26_ = −2.19, *p* = 0.0072) (Figure 8b) and epiphyte density (*b* = −0.0570, *t*_26_ = −5.437, *p* = 0.00001), both negative associations and all significant. The results were very similar for Shannon diversity (conditional model averages: herbs *b* = −0.436, *p* = 0.0044; small trees *b* = 0.0825, *p* = 0.0433; epiphytes *b* = −0.121, *p* < 0.0001, Figure 8c), although in addition, the best model suite contained liana density, a positive but not significant association (conditional model average *b* = 0.0630, *p* = 0.120). NMDS1 showed, in the best suite of two models, positive associations with bamboo cover (conditional model average *b* = 0.452, *p* < 0.0001) (Figure 8d), epiphyte density (conditional model average *b* = 0.0612, *p* = 0.0031), and tree fern density (conditional model average *b* = 1.084, *p* = 0.186), and a negative association with liana density (conditional model average *b* = −0.0799, *p* = 0.0324).

## 4. Discussion

This study aimed to increase our understanding of the factors affecting Darwin wasp diversity in tropical ecosystems, which might ultimately aid conservation of this very diverse and functionally important group of insects. By collecting pimpline Ichneumonidae along a transect up a tropical mountain, we sought to determine (a) whether species richness was high relative to other equivalent collections, both in tropical and more temperate regions; (b) how the community varied along the elevation gradient, and in particular, how alpha and beta diversity were affected by elevation; and (c) whether any other features of the habitat could predict changes in the Darwin wasp community. Our study found that (a) richness was high overall compared with similar studies of the same group, especially at our low- to medium-elevation sites (but not at high elevations); (b) different community metrics responded differently to elevation, with species composition being strongly affected, leading to species turnover. Abundance, richness and diversity were always low at high elevations, although only diversity was high at low elevations, with richness and abundance peaking at (different) intermediate elevations; and (c) vegetation generally does not predict changes in the pimpline wasp community above and beyond elevation, although several aspects of vegetation are correlated with the elevational changes in the pimpline wasp community. Below, we consider these results in the context of other work and discuss their significance for the ecology and conservation of pimplines, and more generally, for parasitoid wasps.

### 4.1. Effect of Latitude

Some data imply, at face value, that there might be an “inverse” relationship between latitude and species richness in Ichneumonidae, with richness peaking at intermediate latitudes [41,42,91]. These data have spawned a plethora of studies, with some investigating the biological basis of such a trend [43], and others investigating its veracity [40,45,47,48,109]. In particular, there is doubt about whether estimates of Ichneumonidae richness in the tropics are based on sufficiently complete and accurate samples. One potential inaccuracy is implied by molecular genetic studies, which have sometimes identified large numbers of cryptic species in parasitoid wasps [47,48]. Like the vast majority of previous studies, our study was based purely on morphological characterization of species; therefore, it does not address this possibility.

However, a second possibility is that, because tropical species abundance distributions are flatter [110], many species are found in low abundance. Consequently, greater sampling intensity is required to accurately estimate species richness. Some recent studies that have conducted intensive sampling in tropical or subtropical regions have indeed recorded high numbers of species [42,45,48]. Furthermore, when accounting for sampling intensity, tropical collections of pimpline wasps generally display higher richness than equivalent non-tropical studies, to the extent that a simple control for sampling intensity across studies can reveal a typical latitudinal richness gradient [45]. In our study, we sampled 1560 individuals in 180 Malaise trap months. Only six of the 97 collections in [45] sampled more individuals, and only three sampled for more Malaise trap months. Thus, our sampling was relatively intensive and could potentially reveal useful information about the latitudinal richness gradient.

Our study found 98 morphospecies. This is higher than all but one of the pimpline Malaise collections in [45] (which only found seven more species from about the same number of individuals but 240 Malaise trap months, and sampled Rhyssinae as well as Pimplinae), suggesting that it is rare to find such richness in a single study from a single locality. A recent study, [63], also from SE Brazil, found almost as many species but from only half as many individuals sampled, implying even higher richness than we found. One potential issue with this simplistic comparison is that the studies in [45] were from a narrow range of elevations (generally < 200 m). If taxonomic composition generally changes with elevation, which has been found in some studies [55,61,62,63], including ours (see below), this could explain the higher than expected richness in our study. To make our data more comparable with these, we split them into seven elevational zones, all less than or close to 200 m in elevation range coverage. Investigation of rarefaction curves for these different zones confirmed that for low and intermediate elevations, richness was generally higher than in the literature dataset, although for the highest elevation zones, it was more consistent with the bulk of the literature dataset. Thus, the high number of species we found overall is not solely due to elevational turnover (see below) but is also due to high alpha diversity within individual elevational zones. The taxonomic expertise employed on our study probably facilitated the identification of a large number of morphospecies. Although many equivalent studies have employed similar expertise, across the larger sample of studies [45], decisions are likely to involve some degree of subjectivity and account for some of the variation across studies.

Our study further suggests that even with a relatively high number of species observed, and high sampling intensity, the community was under-sampled and that 12–46% more species, possibly more, lie undiscovered, awaiting more sampling. The existence of under-sampling is typical in Malaise trap sampling of parasitoids [44,45,65,76,94], further exacerbating uncertainty about latitudinal richness gradients. Complementary sampling through other techniques would likely be necessary to provide a more complete inventory of species [91]. Though our study contributes only one or a small number of datapoints to the overall literature (depending on whether and how the data are split), it tends to support a more typical latitudinal richness gradient in Pimplinae in this respect. To give one crude but illustrative comparison, the number of pimpline species found on this one mountain using one sampling technique over 6 months is equivalent to the number ever found in the whole of the British Isles (109 species [111]). The extent to which Ichneumonidae in general conform to this pattern will depend on the extent to which other subfamilies of Ichneumonidae differ from Pimplinae. Some data do suggest that different subfamilies of ichneumonids dominate richness to different extents in different regions, and some subfamilies, such as Diplazontinae, Tryphoninae, and Ctenopelmatinae, are unlikely to show high tropical richness because their hosts are largely temperate groups [91]. If, however, pimplines were generally representative, the implications for conservation would be that tropical forest environments are more important for ichneumonids than an “inverse” gradient would tend to imply. Moreover, within the tropics, mountainous zones with a high elevational range may harbor more species in this group and could be good targets for conservation efforts, such as protection or restoration. Data already suggest that the also diverse braconid wasps and some other parasitoids are likely to show a more typical latitudinal richness gradient, with highest richness in the moist tropics [91,112].

### 4.2. Effect of Elevation and Temperature

Previous studies of how parasitoid communities vary with elevation have produced a range of different patterns, with species richness sometimes peaking at intermediate elevations [53,54,55], decreasing with elevation in some taxa [55], increasing with elevation [57,58,59], or showing no overall trend [56]. In addition, some studies have shown that abundance, richness, and diversity can show contrasting patterns with elevation [54,58], but this is not ubiquitous [55]. Furthermore, different taxonomic groups in the same collections have been shown to display contrasting patterns of richness with elevation [55,61,62,63]. This typically results in turnover of species with elevation. These varying patterns can have a significant impact on conservation strategies. This is because they can determine which elevations have the highest standing diversity, and might be beneficially protected, but also to what extent extending the range of protected elevations brings added benefits in terms of protected biodiversity.

Some previous studies have included a relatively small number of elevational sites, meaning that they cannot detect complex patterns [44,63]. Others include sites that can be distantly separated [55,60], whilst sites can be sampled with varying intensities [55]. The strengths of our study are that we included 15 elevational sites, meaning that we could detect complex patterns of community variations with elevation. Our study was well replicated, with 30 Malaise traps overall, two at each site (meaning that traps in untypical sites have little influence on the results). Moreover, our study sites were not widely separated geographically but were located on the side of a single mountain. This meant that variation with elevation was less likely to be confounded by other geographic factors. All our sites were sampled with the same intensity (in terms of Malaise trap months), meaning that we can be more confident that sampling intensity did not control the observed patterns.

Taken at face value, our data demonstrate that abundance, richness, and diversity show contrasting patterns with elevation. If conservation strategies were constrained to just a single elevational site, then the choice of site would be the lowest site to maximize Simpson’s Index of Diversity, slightly higher to maximize Shannon diversity, higher still to maximize species richness, and even higher to maximize overall pimpline abundance, although the highest elevations would never be optimal from any of these perspectives. However, if conserving a range of elevations becomes possible, the optimal strategy is not so intuitively obvious because our data additionally demonstrate turnover in species composition with elevation. This could potentially mean that higher elevations would be included in an optimal set of protected sites, depending on the elevational ranges and precise degree of species turnover across sites. Explicit consideration of the elevational range variation of species across our transect, and explicit reserve selection simulations are needed to confirm whether this is the case [77]. The elevational patterns in our data are similar to many commonly found in other charismatic taxa that usually form the basis of conservation planning [51]. This, in turn, would suggest that pimplines may generally be well served by conservation decisions based on other taxa, although precise site-specific comparisons, beyond the scope of this study, would be important. A caveat is that in the Atlantic Rainforest, and likely some other tropical mountains, there is a high proportion of botanical endemism in the highest elevation environments [81]. Conserving this botanical diversity is obviously important, but our data provide little indication that this would enhance conservation of pimpline wasps as a by-product, since this area had the lowest standing richness. However, this may change as climate and elevational ranges change [62,88].

Some issues with our data require appropriate interpretative caution. One is that we only sampled with a single, albeit widely used and efficient technique: Malaise traps at ground level. A large number of studies have shown that different collection techniques often produce different results in ichneumonid and other insect collections, indicating that all have biases [53,113,114,115]. At present, we have no way of knowing how our findings might change with a broader suite of collecting methods. Time and effort are the main barriers to implementing this. Another issue is that our sample sizes for wasps at high elevation sites are very small, presumably partly due to reduced flight activity at lower temperatures. Given that the rarefaction curve for the higher elevation sites has not asymptoted (Figure 3d), further sampling at the higher elevation sites would probably accumulate more species, and only doing so could confirm to what extent. Other issues are that in this study, we only included 6 months of pimpline collection, only sampled during a single year. It is possible that our samples are not typical of year-to-year variations since insect populations often fluctuate considerably [91].

Given strong patterns in community variation with elevation, a natural question is what controls this variation. The fact that communities are relatively depauperate at high elevation could be controlled by a number of physical, ecological, or evolutionary factors [50], such as the reduction in habitat area with elevation; the decline in productivity, leading to reduced energy flow, reduced population size, and higher extinction rates; and the need for special, cold-adapted physiology, which may only rarely evolve in tropical systems [51,60,116,117,118,119]. Perhaps less intuitive is that the lowest elevations do not always contain the greatest number of individuals or species [51]. In our case, one possible contributing factor is that anthropogenic disturbance is highest at lower elevations [120] and tends to significantly reduce biodiversity in this region [121]. Alternative factors might include the mid-domain effect, whereby there is more overlap of species’ elevational ranges at mid elevations, leading to greater richness [116,119]; ecotone effects, such as reduced habitat heterogeneity at low elevations, leading to reduced ecological niche segregation [50]; and in Ichneumonidae, a range of proposed explanations (e.g., the “nasty host” hypothesis) for an “inverse” latitudinal richness gradient, which could also operate over elevational gradients [43,91].

In pimplines, as for parasitoids more generally, host availability is a likely important intermediary that controls community patterns. Some previous studies on parasitoid community patterns have focused on direct host sampling [35,47,122]. However, it is unlikely that host data will ever be a helpful conservation surrogate in very rich groups because it lacks practicality, and for that reason, we have focused our efforts on different explanatory variables: abiotic factors and broad vegetation changes. Follow-up studies looking at proximate factors might focus on whether potential hosts follow similar diversity patterns to their parasitoids, although precise host usage data (unavailable for most of our species) for the community would be useful to design such a study. There are also numerous reasons why broad parasitoid and host community patterns might diverge, such as plasticity in host use in different environments [123].

The fact that the pimpline community varies with elevation suggests that temperature is likely to be an important ultimate controlling factor [50]. In our study, as elsewhere around the world, elevation strongly predicts temperature, such that they can be extremely good proxies for each other. Given about 2000 m in elevational range considered in our study, and the typical 0.5 °C drop in mean annual temperature with a 100 m elevation increase [88], our study covers about 10 °C range in mean (plus maximum and minimum) monthly temperatures, with the mean 22 °C at the bottom and 12 °C at the top of the transect. It is interesting to speculate what this might imply globally if temperature were taken as a good proxy for pimplines elsewhere in the world, although there are many reasons why local-scale alpha richness patterns might not translate to larger scales, particularly because of variation in beta diversity. The covered temperature range is typical of the means of low latitude temperate regions, subtropical regions, and some higher latitude tropical regions [124]. Using our data as a proxy, this would suggest that abundance and richness peaks in this region rather than nearer the equator, but that Simpson’s diversity could still be maximal at the equator if the model were extrapolated. It would be interesting to document patterns of beta diversity in Ichneumonidae over larger spatial scales, and to study a local community over a wider range of elevations so as to cover a larger temperature gradient. A high equatorial mountain would be needed for this.

### 4.3. Effects of Vegetation

Previous studies have shown that parasitoid wasp communities can sometimes vary across vegetation or habitat types [45,64,65,66,67], or other features of the habitat, such as the richness of some plant groups [64,65,66,72,73,74,75,76] and structural complexity [57,68,69,70,71]. In principle, these could serve as proxies for parasitoids, to be used in conservation planning or to aid optimal management of existing habitat [77]. In our study, we therefore measured several features of the habitat and vegetation to see if they might predict some of the variation in the pimpline community. We wished to document some of the main functional and structural changes that occur in the vegetation across our trapping sites; therefore, we documented changes in the cover of functional groups (herbs, ferns, bamboo) and mass and moisture content of leaf litter at ground level, as well as other components of the vegetation away from ground level (the density of trees, palms, tree ferns, lianas, and epiphytes). PCA confirmed that most of these variables are important contributors to habitat variation.

Correlation matrices and PCA showed that many of these variables also predict community composition of pimplines, and indeed, there are also some vegetation variables that are not well predicted by elevation alone (orthogonal to it). However, linear models showed that only the first PC of habitat variables predicted community properties consistently, although the further components also contributed to explaining community composition. The first component is heavily weighted by variables that closely correlate with elevation, including temperature, litter moisture, bamboo and fern ground cover, and epiphyte density. Since all of the above vegetation variables are positively correlated with elevation, these are variables that are associated with less rich and diverse communities, and if their influence was direct and causal, they would have to exert this by suppressing diversity, perhaps because they provide little diversity of hosts for parasitoids compared with other vegetation types.

Alternatively, these vegetation variables could merely be passively and indirectly correlated with the pimpline community because they are themselves correlated with other, third, variables which also are correlated with it. A likely candidate is temperature. Indeed, when linear models incorporating vegetation variables were constructed, which included elevation (almost perfectly negatively correlated with temperature), the most common outcome was that nothing else was a significant predictor. When only vegetation variables were included, a number of them could be significant predictors: these included negative effects of bamboo cover, epiphyte density, and herb cover again, but also positive effects of small tree density and liana density, which might plausibly increase diversity or richness by increasing host species richness.

Thus, our data tentatively suggest that managing tropical forests to increase the density of small trees and lianas (and less plausibly reducing herb, bamboo, and epiphyte cover/density) might improve richness and diversity of some parasitoids, although further (e.g., experimental) work is needed to confirm whether these relationships are causal. These and other vegetation variables might act as useful proxies for the parasitoid community in reserve selection or other conservation-management tasks, but elevation and temperature are probably more effective proxies and much more practical to measure.

## 5. Conclusions

Our study provides further evidence for high tropical richness of pimpline wasps through intensive sampling. It provides strong evidence that alpha diversity and richness are lowest at high elevations and peak at low to intermediate elevations at least in one locality, but that high species turnover can occur across elevations. The study also provides some potential vegetation surrogates for richness and diversity, notably, the density of small trees. Further studies could address the ecological causes of these patterns and determine how these findings could be translated into effective conservation tools for pimpline wasps.

## Figures and Tables

**Figure 1 insects-14-00861-f001:**
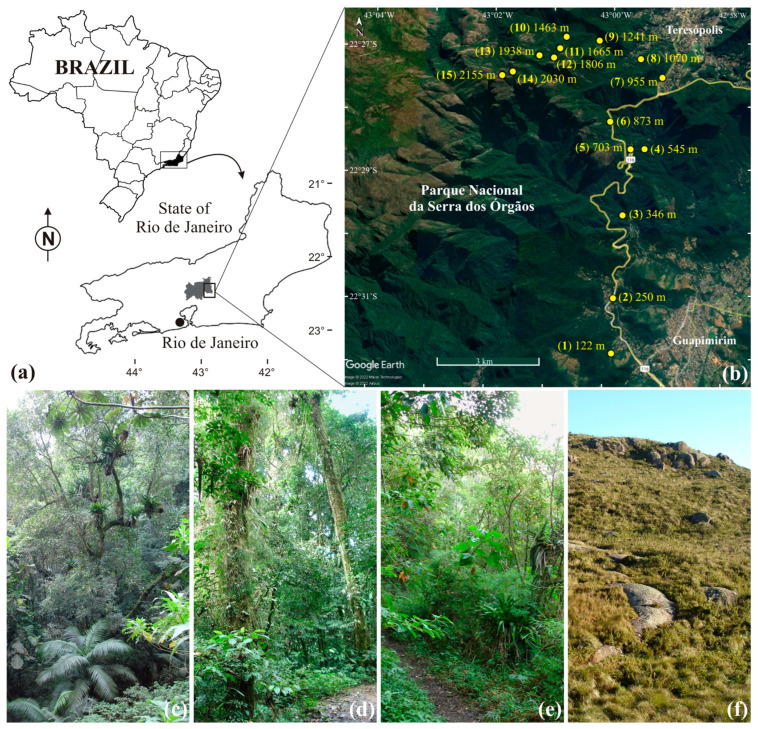
(**a**) Location of the Serra dos Órgãos National Park (gray shaded area), a protected tropical Atlantic Rain Forest in the State of Rio de Janeiro, southeast Brazil. (**b**) Map of the 15 study sites and their respective altitudes along the elevational gradient in the Park, where four different phytophysiognomies are observed: (**c**) lower montane forest (up to 500 m), which presents a 20 m high canopy but normally no other well-defined forest layers; (**d**) montane forest (500 m to 1500 m), with its clear stratification into arboreal, shrub, and herb layers, and large emergent trees reaching 40 m covered with abundant lianas and epiphytes; (**e**) high montane forest (1500 m to 2000 m) with smaller trees of up to 10 m covered with mosses and epiphytes, and great diversity of shrubs; and (**f**) high-altitude grassland, also known as campos de altitude (above 2000 m), dominated by herbal vegetation growing around rocks and scattered shrubs.

**Figure 2 insects-14-00861-f002:**
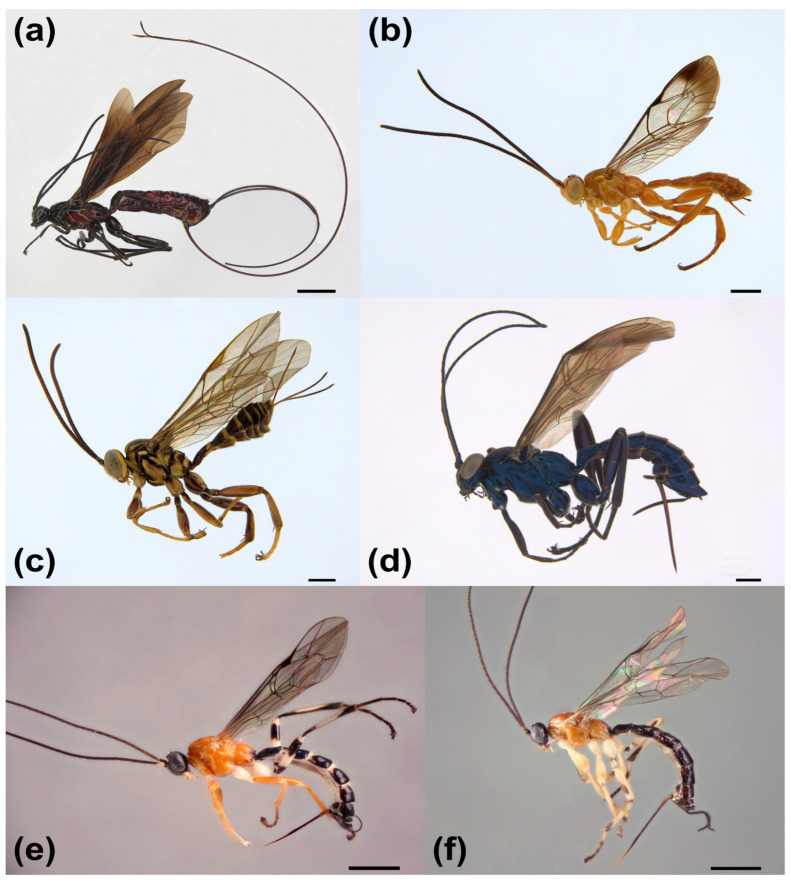
Some of the Pimplinae wasp species sampled (all are females). (**a**) *Dolichomitus megalourus* (scale bar 4 mm), 10 individuals sampled; (**b**) *Neotheronia charli* (scale bar 1 mm), 24 individuals sampled; (**c**) *Neotheronia* sp. 6 (scale bar 1 mm), 26 individuals sampled; (**d**) *Pimpla caerulea* (scale bar 1 mm), 447 individuals sampled; (**e**) *Polysphincta organensis* (scale bar 2 mm), 19 individuals sampled; and (**f**) *Polysphincta teresa* (scale bar 2 mm), 8 individuals sampled.

**Figure 3 insects-14-00861-f003:**
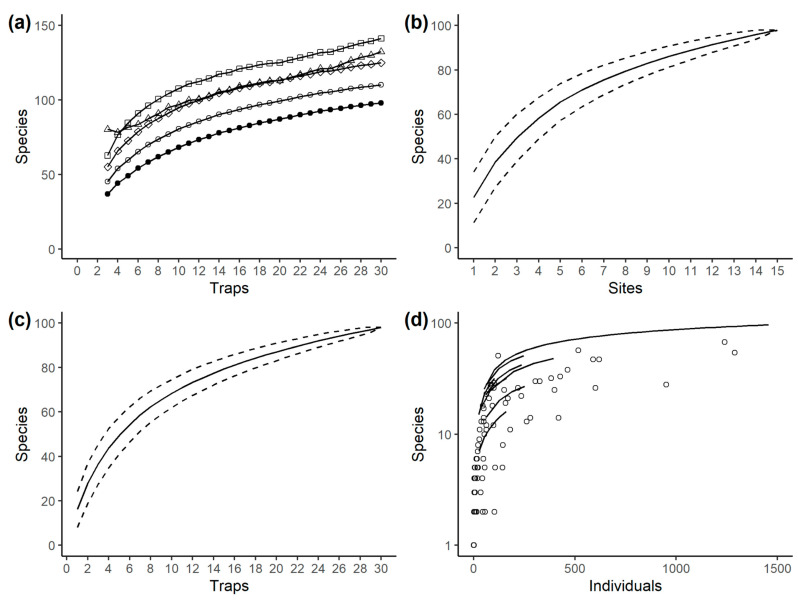
Species richness of pimplines against sampling intensity. (**a**) Estimates of total species richness against number of traps sampled (filled circles: observed data; open circles: bootstrap; diamonds: first-order jackknife; triangles: Chao; squares: second-order jackknife); (**b**) site-level rarefaction (±SD); (**c**) trap-level rarefaction (±SD); (**d**) mean individual-level rarefaction for the whole data (top line) and altitudinally-zoned subsets (from top to bottom, 332–549 m, 703–887 m, 952–1071 m, 110–150 m, and 1236–1482 m, which are superimposed, 1649–1812 m and 1935–2169 m), points are literature-based data covering the same span of sampling intensities for comparison from [45]. Note the log scale on the *y*-axis.

**Figure 4 insects-14-00861-f004:**
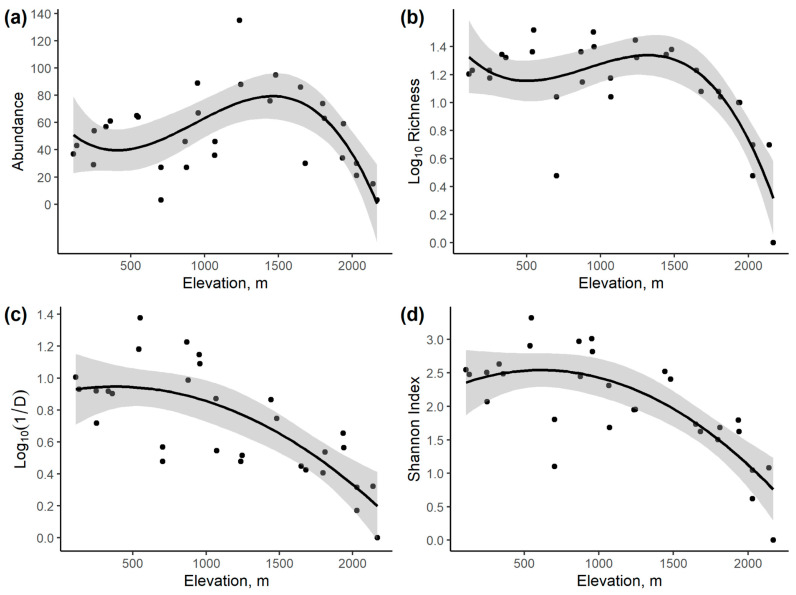
Pimplinae wasp community metrics against elevation (altitude) across 30 traps. (**a**) Abundance; (**b**) Log_10_ Species Richness; (**c**) Log_10_ Simpson’s Index (1/*D*); and (**d**) Shannon Index. Lines are the equations of the polynomial linear model in Table 2 with the lowest AICc, ±95%CI; (**a**,**b**): cubic models (**c**,**d**): quadratic models.

**Figure 5 insects-14-00861-f005:**
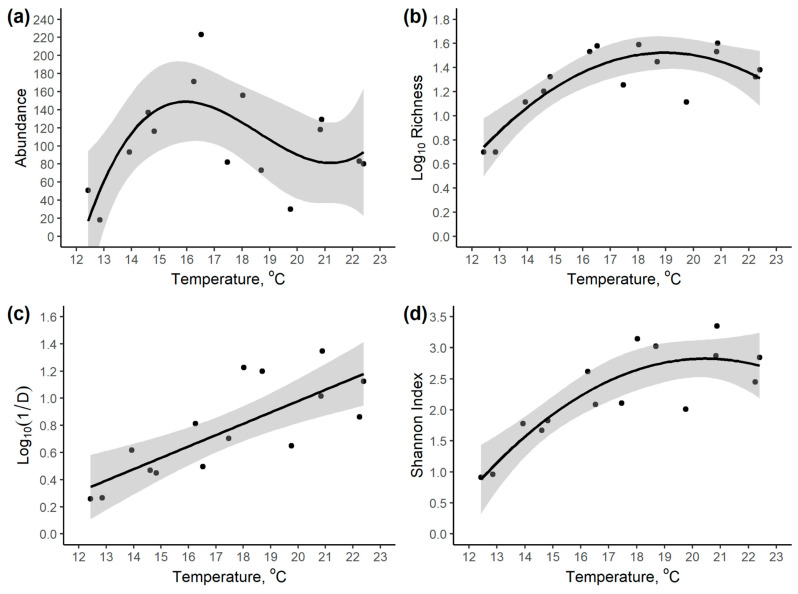
Pimplinae wasp community metrics against mean monthly temperature across 15 sites. (**a**) Abundance; (**b**) Log_10_ Species Richness; (**c**) Log_10_ Simpson’s Index (1/*D*); and (**d**) Shannon Index. Lines are the equations of the model in Table 3 ± 95%CI. (**a**) cubic model; (**b**,**d**) quadratic models; and (**c**): linear model.

**Figure 6 insects-14-00861-f006:**
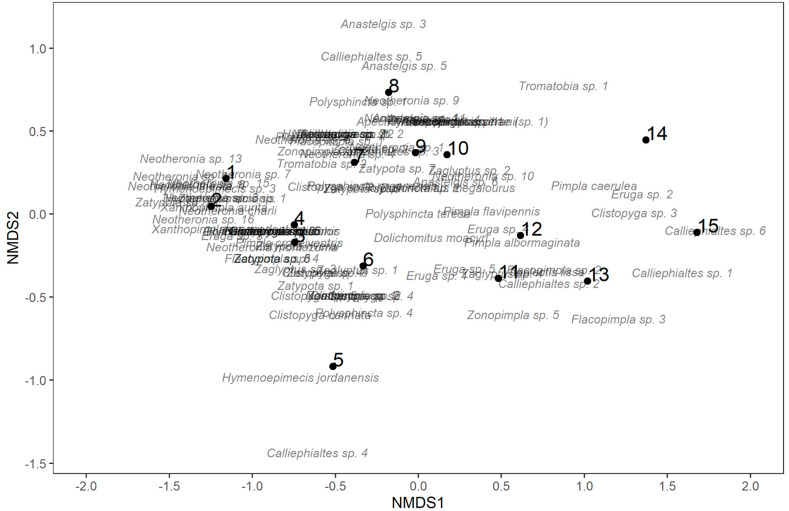
An ordination using Non-metric Multidimensional Scaling (NMDS) of the pimpline community at the site level. Black numbers and points indicate the 15 sampling sites, going from the bottom of the mountain (1) to the top (15). Species are in gray, small lettering.

**Figure 7 insects-14-00861-f007:**
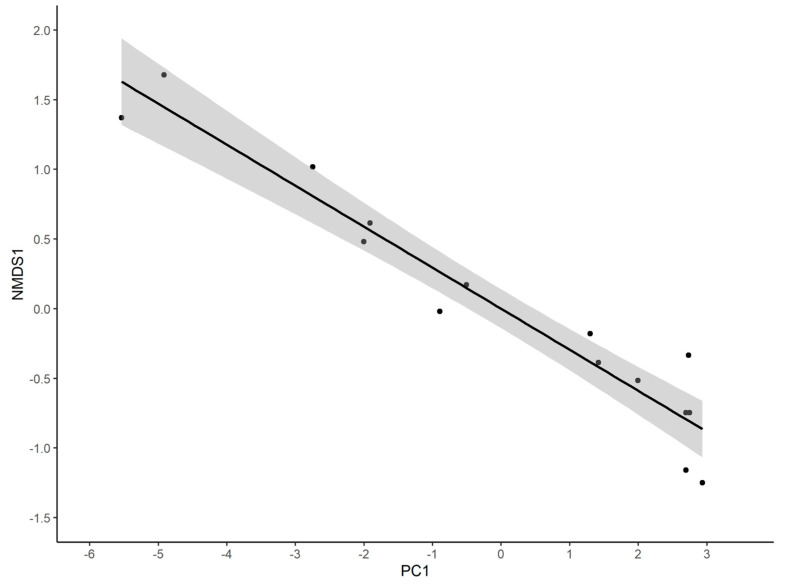
Pimplinae community composition, as measured by the first axis of a Non-Metric Multidimensional Scaling analysis (NMDS1, see Figure 6) across sampling sites (*n* = 15), against the first Principal Component (PC1) of the habitat variables at those sites (see Table 4). The line is the linear regression (±95% CI in gray). The figure demonstrates that pimpline community composition is very strongly associated with differences in habitat characteristics across sites.

**Figure 8 insects-14-00861-f008:**
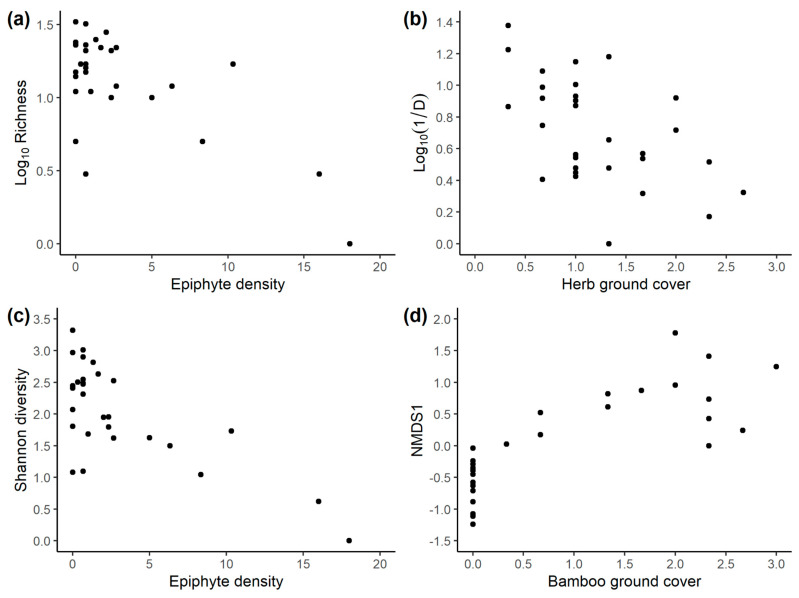
Vegetation predictors of the pimpline community across traps (*n* = 30). (**a**) Epiphyte density against log richness; (**b**) herb ground cover against the log of inverse Simpson’s Index of Diversity; (**c**) epiphyte density against Shannon diversity; (**d**) bamboo ground cover against the first axis of a Non-metric Multidimensional Scaling analysis of pimpline community composition. For sampling units, see Methods text.

**Table 1 insects-14-00861-t001:** Incidence (Chao, first-order jackknife, second-order jackknife, bootstrap) and abundance (bias-corrected Chao and ACE) based estimators (±SE) of the total pimpline species pool from which we sampled. The bias-corrected Chao and ACE estimators are based on abundance across all samples combined. Standard errors are not available for second-order jackknife.

Method	Trap Level Data (*n* = 30)	Site Level Data (*n* = 15)
Observed richness	98	98
Chao	132 ± 18	132 ± 17
First-order jackknife	125 ± 8	127 ± 11
Second-order jackknife	141	143
Bootstrap	110 ± 5	111 ± 6
Bias-corrected Chao	123 ± 13	
ACE	122 ± 6	

**Table 2 insects-14-00861-t002:** Linear models of pimpline community metrics (response variable) against elevation at trap level (*n* = 30). Model coefficients are given for the predictor variables, * significant predictor. Models within two AICc units of the model with the lowest AICc are in bold.

Response Variable	Intercept	Elevation	Elevation^2^	Elevation^3^	Elevation^4^	Model *r*^2^	AICc
Abundance	54.90 *	−0.00254	-	-	-	0.003	294.05
	19.60	0.0882 *	−3.946 × 10^−5^ *	-	-	0.238	288.66
	**62.10 ***	**−0.1217**	**−1.912 × 10^−4^ ***	**−6.835 × 10^−8^ ***	-	**0.424**	**283.20**
	55.75	−7.606 × 10^−2^	1.073 × 10^−4^	−1.270 × 10^−8^	−1.211 × 10^−11^	0.425	286.26
Log_10_ Species Richness	1.423 *	−2.723 × 10^−4^ *				0.255	16.79
	1.032 *	7.323 × 10^−4^ *	–4.369 × 10^−7^ *			0.495	8.835
	**1.456 ***	**−1.362 × 10^−3^**	**1.866 × 10^−6^ ***	**−6.822 × 10^−10^ ***		**0.633**	**2.196**
	**1.228**	**2.745 × 10^−4^**	**−1.145 × 10^−6^**	**1.315 × 10^−9^**	**−4.344 × 10^−13^**	**0.649**	**3.691**
Log_10_ Inverse Simpson’s Index (1/*D*)	**1.122 ***	**−3.599 × 10^−4^ ***				**0.512**	**3.85**
	**0.9136 ***	**1.765 × 10^−4^**	**−2.333 × 10^−7^**			**0.576**	**2.31**
	0.8901 *	2.925 × 10^−4^	−3.608 × 10^−7^	3.778 × 10^−11^		0.576	5.18
	0.7449 *	1.3 × 10^−3^	−2.213 × 10^−6^	1.267 × 10^−9^	−2.673 × 10^−13^	0.583	7.87
Shannon Index	2.925 *	−0.000790 *				0.479	54.93
	**2.268 ***	**8.973 × 10^−4^**	**−7.339 × 10^−7^ ***			**0.602**	**49.53**
	**2.643 ***	**−9.526 × 10^−4^**	**1.3 × 10^−6^**	**−6.025 × 10^−10^**		**0.624**	**50.74**
	2.266 *	1.756 × 10^−3^	−3.863 × 10^−6^	2.702 × 10^−9^	−7.19 × 10^−13^	0.633	53.15

**Table 3 insects-14-00861-t003:** Linear model coefficients of mean monthly temperature against pimpline community properties at the site level (*n* = 15). Only the models with the lowest AICc scores are shown. * significant predictor.

Response Variable	Intercept	Temperature	Temperature^2^	Temperature^3^	Model *R*^2^
Abundance	−5497.70	+945.02	−51.91	+0.9314	0.458
Log_10_ Richness	1.221 *	7.036 × 10^−4^ *	−4.225 × 10^−7^ *	-	0.670
Log_10_ (1/*D*)	−0.6927	+0.0836 *	-	-	0.614
Shannon Index	−9.803 *	+1.234 *	−0.0302 *	-	0.769

**Table 4 insects-14-00861-t004:** Principal component weightings for the abiotic variables at site level (*n* = 15), along with initial eigenvalues and % variance explained. For variable explanations and units, see Methods text. Variables were scaled prior to analysis.

Variable	PC1	PC2	PC3	PC4
Elevation	−0.326	−0.138	−0.090	−0.117
Mean temperature	0.328	0.103	0.133	0.152
Max. temperature	0.303	0.153	0.224	0.182
Min. temperature	0.333	0.084	0.107	0.139
Temperature amplitude	−0.258	0.159	0.290	0.050
Humidity	0.082	0.298	−0.337	−0.362
Dry litter mass	0.091	−0.541	−0.020	−0.012
Litter moisture	−0.298	0.071	−0.207	−0.078
Large tree density	0.028	−0.413	−0.456	0.152
Small tree density	−0.087	0.133	−0.385	0.420
Tree fern density	0.113	0.179	−0.047	−0.657
Bamboo ground cover	−0.326	−0.078	0.041	0.050
Fern ground cover	−0.290	0.197	−0.080	−0.020
Herb ground cover	−0.174	0.430	0.058	0.289
Epiphyte density	−0.317	0.085	0.011	0.073
Liana density	0.189	0.216	−0.445	0.200
Palm density	0.189	0.156	−0.324	−0.088
Initial eigenvalue	8.354	2.657	1.804	1.476
% variance	49.14	15.63	10.61	8.69
% cumulative variance	49.14	64.78	75.39	84.07

**Table 5 insects-14-00861-t005:** Best models of pimpline community properties against the abiotic variable Principal Components (PC) in Table 4. Best models were selected on the basis of AICc scores. * significant predictor.

Response Variable	Intercept	Predictor Variables		Model *R*^2^
Abundance	145.75 *	−6.80 × PC1	−5.35 × PC1^2^ *	0.467
Log_10_ richness	1.444 *	+0.040 × PC1	−0.019 × PC1^2^ *	0.745
Log_10_(1/*D*)	0.766 *	+0.104 × PC1 *		0.719
Shannon Index	2.244 *	+0.227 × PC1 *		0.772

## Data Availability

Data are in the Appendix A.

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
