# Peer review of "Variation in a Darwin Wasp (Hymenoptera: Ichneumonidae) Community along an Elevation Gradient in a Tropical Biodiversity Hotspot: Implications for Ecology and Conservation"

_insects, 2023, doi:10.3390/insects14110861_

Round 1

Reviewer 1 Report

Comments and Suggestions for Authors

The subject manuscript is very thorough in all respects. I detect no issues that should hinder publication. It should be noted that I am not a statistician, and I am not particularly well-versed in the use of diversity measures and indicies.

I do have one question that the authors may choose to address: It is noted that the species accumulation curves had not quite reached their asymptotes. Is it possible that higher-elevation sites could “catch up” to lower elevation sites if given longer sampling times? Presumably, while the trapping time was the same, the number of degree days suitable for insect activity would have been lower at high-elevation sites due to cooler temperatures overall, and possibly a more dramatic diurnal change. This effect would be compounded by the apparent difficulty with developing cold-tolerance in tropical ecosystems, as cited by the authors.

Overall, a very nice paper and worthy contribution to the literature.

Author Response

Referee 1

I do have one question that the authors may choose to address: It is noted that the species accumulation curves had not quite reached their asymptotes. Is it possible that higher-elevation sites could “catch up” to lower elevation sites if given longer sampling times? Presumably, while the trapping time was the same, the number of degree days suitable for insect activity would have been lower at high-elevation sites due to cooler temperatures overall, and possibly a more dramatic diurnal change. This effect would be compounded by the apparent difficulty with developing cold-tolerance in tropical ecosystems, as cited by the authors.

Response: This is an excellent point. In Figure 3d we have shown that the higher-elevation sites have lower richness than other sites for equivalent sampling, but the referee is correct that we do not know what richness would be expected at those sites with more sampling, and due to low sample size extrapolation methods will be inconclusive. We have added some text to the discussion on page 26, para 2 to make this point so that readers can use due caution when interpreting the results. 

Reviewer 2 Report

Comments and Suggestions for Authors

Given the extreme detail of the paper, I use this space to address several fundamental questions that require attention.

24, par. 1 -Can pimplines speak for ichneumonids, parasitoids in general? What about braconids?

24, par. 2 – These other studies generally focus on ichneumonids, not specially on one of their subgroups (here, pimplines).  Do the comparisons hold when considering a subgroup against the family?

24, par. 5 – More consideration on the well-known bias from the sole use of Malaise traps is needed.

25, par. 1 – The equivalence of pimplines and ichneumonids presented here needs attention. I think that the relationship between diversity, etc. in pimplines and ichneumonids in general, and braconids, needs more attention. Also, 26, par. 2.

25, par. 2 – With the attention paid to conservation-related issues, to what extent do your results resemble those reported for groups more traditionally considered in this regard (charismatic species, etc., etc.)?

27, par. 2 –What roles do host diversity, abundance, etc. play in this system? It seems that they have been largely ignored in this study. Obtaining the results you sought would only be possible if there were adequate hosts. The word “host” only appears in passing. You need to address this issue.

Author Response

Referee 2. 

24, par. 1 -Can pimplines speak for ichneumonids, parasitoids in general? What about braconids?

Response: This is an interesting point, also repeated below. Since pimplines are members of the Ichneumonidae and also are parasitoid wasps, the data are relevant to those broader taxa/guilds, and in, say, a meta-analysis of the broader taxa/guild would qualify for inclusion. It remains the case that we do not know, without further e.g. a meta-analysis, or more inclusive original sampling, how representative they might be, and we presume that this is the referee’s point, which is valid and correct. Other work does suggest, in confirmation of the referee’s point, that some groups of Ichneumonidae display different richness patterns to pimplines, and different families also seem to show different richness patterns. We have changed the phrase page 24 para 1 “Darwin wasp” to “pimpline wasp” here and elsewhere in the discussion since that more accurately reflects our data, and removed the phrase “parasitoid wasps” in the next paragraph. We retained the phrase parasitoid wasps in some places since our discussion is relevant to the wider guild. We have added some text later in the discussion (page 25, para. 1) to highlight this issue to readers so that they may apply appropriate caution.

24, par. 2 – These other studies generally focus on ichneumonids, not specially on one of their subgroups (here, pimplines).  Do the comparisons hold when considering a subgroup against the family?

Response: Our response above is also relevant here. The study by Gomez et al. in the reference list looks at pimplines specifically across a large number of similar studies and finds some evidence in favour of a decline in richness with latitude once sample size is accounted for, although other studies generally do not account for this and other confounding variables. In common with our points above, we think that patterns in pimplines are relevant to patterns in the Ichneumonidae, but of course need not conform to them for the general Ichneumonidae pattern to be different. Other work does suggest, in confirmation of the referee’s point, that some groups of Ichneumonidae display different richness patterns to pimplines. As stated above, we make this point  now on page 25, para 1. 

24, par. 5 – More consideration on the well-known bias from the sole use of Malaise traps is needed.

Response: this is a fair point and we are happy to comply. We now include discussion of this and references to work that compares different sampling methods, p.26 para 2. 

25, par. 1 – The equivalence of pimplines and ichneumonids presented here needs attention. I think that the relationship between diversity, etc. in pimplines and ichneumonids in general, and braconids, needs more attention. Also, 26, par. 2.

Response: we now include explicit reference to braconids on p. 25 para 1 and the same paragraph already raises the issue of taxonomic equivalence. 

25, par. 2 – With the attention paid to conservation-related issues, to what extent do your results resemble those reported for groups more traditionally considered in this regard (charismatic species, etc., etc.)?

Response: That’s an excellent point. We now include reference to broader studies on other taxa, page 26, para 1.

27, par. 2 –What roles do host diversity, abundance, etc. play in this system? It seems that they have been largely ignored in this study. Obtaining the results you sought would only be possible if there were adequate hosts. The word “host” only appears in passing. You need to address this issue.

Response: This is also an excellent point. We agree that hosts would seem a necessary precondition to support whatever communities of pimplines exist, but are also probably not sufficient to fully explain what exists, as other factors will likely also impinge separately or in combination. Some studies of parasitoid communities engage in explicit host sampling, although they are usually then more taxonomically confined as a result of needing to focus on a restricted range of hosts. Global meta-analyses of the parasitoids attacking particular hosts (such as those by Hawkins et al.) are useful but sample only a part of any  local wasp community. The issue of hosts is interesting with regard to determining causality of patterns, but of potentially less use from a conservation surrogacy perspective as hosts are likely to be more time-consuming to survey than vegetation and abiotic variables. We now include discussion of hosts in this context on page 27 para1.

Other changes.

We have taken the opportunity to make a few small changes elsewhere: 

  1. 4. A small addition to the site information in the methods section.
  2. 24. Small changes to the text in para 3 and 4.
  3. 25, 1st para. Two extra sentences at the top.
  4. 26 para 2: broadened the discussion of caveats so this is more comprehensive. 

References: extra references added. 

We also realized that the R project in the supplementary materials did not include some small changes to the figures in the original submission, so this has now been updated. 

Round 2

Reviewer 2 Report

Comments and Suggestions for Authors

I am satisfied with the authors' response and revision. I have no further suggestions.